# LATENT OFFLINE DISTRIBUTIONAL ACTOR-CRITIC

## ABSTRACT

Offline reinforcement learning (RL) has emerged as a promising paradigm for real world applications, since it aims to train policies directly from datasets of past interactions with the environment. The past few years, algorithms have been introduced to learn policies from high-dimensional observational states in an offline settings. The general idea of these methods is to encode the environment into a smaller latent space and train policies on the top of this smaller representation. In this paper, we extend this general method to stochastic environments (i.e. where the reward function is stochastic) and considering a risk measure instead of the classical expected return. First, we show that under some assumptions it is equivalent to minimize a risk measure in the latent space and in the natural space. Based on this result, we present Latent Offline Distributional Actor-Critic (LO-DAC), an algorithm which is able to train policies in high-dimensional stochastic and offline settings to minimize a given risk measure. Empirically, we show that using LODAC to minimize Conditional Value-at-Risk (CVaR), outperforms previous methods in term of CVaR and return on stochastic environments.

## 1 INTRODUCTION

In a lot of context, human decisions are stored and build interesting datasets. With the successes of modern machine learning tools comes the hope to exploit them to build useful decision helpers. To achieve this, we could use an imitation learning approach (Hussein et al., 2017). But, in this case, we will at best be as good as humans. Moreover, the performance of this approach depends heavily on the quality of the training dataset. In this work we would like to avoid these behaviours and thus, we consider another framework: reinforcement learning (RL). In recent years, RL achieved impressive results in a number of challenging areas, including games (Silver et al., 2016; 2018), robotic control (Gu et al., 2017; Haarnoja et al., 2018) or even healthcare (Shortreed et al., 2011; Wang et al., 2018). In particular, offline RL seems to be really interesting for real world application since its goal is to train agents from a dataset of past interactions with the environment (Deisenroth et al., 2013).

With the digitization of society, more and more features could be used to represent the environment. Unfortunately, classical RL algorithms are not able to work with high-dimensional states. Obviously, we could manually choose a feature subset. However, this choice is not straightforward and it could have a huge impact on the performance. Therefore, it may be be more practical to use RL algorithms capable of learning from high-dimensional states.

It is common to evaluate RL algorithms on deterministic (in the sense that the reward function is deterministic) environment such as the DeepMind Control suite (Tassa et al., 2018). However, in a lot of real world applications, environments are not deterministic but stochastic. Therefore, it might be important to develop algorithms which are able to train policies in these special cases. The motivation of this paper is to provide a method for training policies in an offline settings and working in a high-dimensional stochastic environment.

In this paper, we present Latent Distributional Actor-Critic (LODAC) an algorithm which is able to train policies in a high-dimensional, stochastic environment and in an offline settings. The main idea is to learn a smaller representation and to train the agent directly in this latent space. But instead of considering the expected return, we take into account a risk measure. First, assuming some hypothesis, we show that minimizing this risk measure in the latent space is equivalent to minimize the risk measure directly in the natural state. This theoretical result provides a natural framework : train a latent variable model to encode the natural space into a latent space and then train a policy on

the top of this latent space using a risk-sensitive RL algorithm. In the experimental part, we evaluate our algorithm on high-dimensional stochastic datasets. In the best of our knowledge, we are the first authors to propose an algorithm to train policies in high-dimensional, stochastic and offline settings.

## 2 RELATED WORK

Before going further into our work, we present some related work.

**Offline RL.** Offine RL (Levine et al., 2020) is a particular approach of RL where the goal is to learn policies directly from past interactions with the environment.This is a promising framework for real-world applications since it allows the deployment of already trained policies. Thus, it is not really surprising that offline RL has received a lot of attention the recent years (Wiering & Van Otterlo, 2012; Levine et al., 2020; Yang et al., 2021; Chen et al., 2021b; Liu et al., 2021; Yu et al., 2021a; Wang et al., 2021).

One of the main problem in offline RL is that the Q-function is too optimistic in the case of out-of-distribution (OOD) states-actions pairs for which observational data is limited (Kumar et al. (2019)). Different approaches have been introduced to deal with this problem. For example, there are algorithms that extend the importance sampling method (Nachum et al., 2019; Liu et al., 2019) or extend the dynamic programming approach (Fujimoto et al., 2019; Kumar et al., 2019). Other authors build a conservative estimation of the Q-function for OOD states-actions pairs (Kumar et al., 2020; Yu et al., 2021b). Finally it is also feasible to extend the model-based approach, (Rafailov et al., 2021; Argenson & Dulac-Arnold, 2020).

**Learning with high-dimensional states.** In the previous years, algorithms were proposed to train policies directly from high-states, like images (Lange & Riedmiller, 2010; Levine et al., 2016; Finn & Levine, 2017; Ha & Schmidhuber, 2018; Chen et al., 2021a). Previous work has observed that learning a good representation of the observation is a key point for this type of problem (Shelhamer et al., 2016). In some works, authors use data augmentation techniques to learn the best representation possible (Kostrikov et al., 2020; Laskin et al., 2020; Kipf et al., 2019). But, it has been intensively studied how we should encode the high-dimensional states into the latent space (Nair et al., 2018; Gelada et al., 2019; Watter et al., 2015; Finn et al., 2016). Then, it is a common approach to train the policies using a classical RL algorithm, like Soft Actor-Critic (SAC) (Haarnoja et al., 2018), on the top of this latent representation (Han et al., 2019; Lee et al., 2020). Furthermore, it is also possible to make some planifications in this latent space to improve the performance (Hafner et al., 2019b;a).

**Risk-sensitive RL.** Risk-sensitive RL is a particular approach to safe RL (Garcıa & Fernández, 2015). In safe RL, policies are trained to maximize performance while respecting some safety constraints during training and/or in the deployment. In risk-sensitive RL, instead of maximizing the expectation of the cumulative rewards, we are more interesting in minimizing a measure of the risk induced by the cumulative rewards. Risk-sensitive RL has raised some attention the last few years (Fei et al., 2021; Zhang et al., 2021). Depending on the context, we might consider different risk measures, like Exponential Utility (Rabin, 2013) Cumulative Prospect Theory (Tversky & Kahneman, 1992) or Conditional Value-at-Risk (CVaR) (Rockafellar & Uryasev, 2002). Conditional Value-at-Risk has strong theoretical properties and is quite intuitive (Sarykalin et al., 2008; Artzner et al., 1999). Therefore, CVaR is really popular and has been strongly studied in a context of RL (Chow & Ghavamzadeh, 2014; Chow et al., 2015; Singh et al., 2020; Urpí et al., 2021; Ma et al., 2021; 2020; Ying et al., 2021). Previous work suggest that taking into account Conditional Value-at-Risk instead of the classical expectation, could prevent from the gap of performance between simulation and real world application (Pinto et al., 2017).

## 3 PRELIMINARIES

In this section, we introduce notations and recall concepts we will use later.

**Coherent Risk Measure.** Let $(\Omega, \mathcal{F}, \mathbb{P})$ a probability space and $\mathcal{L}^2 := \mathcal{L}^2(\Omega, \mathcal{F}, \mathbb{P})$. A functional $\mathcal{R} : \mathcal{L}^2 \to (-\infty, +\infty]$ is called a coherent risk measure (Rockafellar, 2007) if

1. $\mathcal{R}(C) = C$ for all constants $C$.

2. $\mathcal{R}((1 - \lambda)X + \lambda X) \leq (1 - \lambda)\mathcal{R}(X) + \lambda\mathcal{R}(X)$ for $\lambda \in (0, 1)$.

3. $\mathcal{R}(X) \leq \mathcal{R})(X')$ when $X \leq X'$.

4. $\mathcal{R}(X) \leq 0$ when $\|X_k - X\|_2 \to 0$ with $\mathcal{R}(X_k) \leq 0$.

5. $\mathcal{R}(\lambda X) = \lambda\mathcal{R}(X)$ for $\lambda > 0$.

Coherent risk measures have some interesting properties. In particular, a risk measure is coherent if and only if, there is a risk envelope $\mathcal{U}$ such that

$$\mathcal{R}(X) = \sup_{\delta \in \mathcal{U}} \mathbb{E}[\delta X] \tag{1}$$

(Artzner et al., 1999; Delbaen, 2002; Rockafellar et al., 2002). A risk envelope is a nonempty convex subset of $\mathcal{P}$ that is closed and where $\mathcal{P} := \{\delta \in \mathcal{L}^2 \mid \delta \geq 0, \mathbb{E}[\delta] = 1\}$.

The definition of a risk measure $\mathcal{R}$ might depends on a probability distribution $p$. In some cases, it may be useful to specify which distribution we are working with. Thus, we sometimes use the notation $\mathcal{R}_p$. There exists a lot of different coherent risk measures, for example the Wang risk measure (Wang, 2000), the entropic Value-at-Risk (Ahmadi-Javid, 2011) or Conditional Value-at-Risk (Rockafellar et al., 2000; Rockafellar & Uryasev, 2002).

Conditional Value-at-Risk (CVaR$_\alpha$) with probability level $\alpha \in (0, 1)$ is defined as

$$\text{CVaR}_\alpha(X) := \min_{t \in \mathbb{R}}\{t + \frac{1}{1 - \alpha}\mathbb{E}[\max\{0, X - t\}]\} \tag{2}$$

Moreover the risk envelope associated to CVaR$_\alpha$, can be written as $\mathcal{U} = \{\delta \in \mathcal{P} \mid \mathbb{E}[\delta] = 1, 0 \leq \delta \leq \frac{1}{\alpha}\}$ (Rockafellar et al., 2006; 2002).

This rigorous definitions may be not really intuitive, but roughly speaking CVaR$_\alpha$ is the expectation of $X$ in the conditional distribution of its upper $\alpha$-tail, and thus is corresponds to the $(1 - \alpha)$ worst case. In this paper, we follow the classical definition of Conditional Value-at-Risk presented in risk measure literature. In particular, $X$ should be interpreted as a loss function.

**Offline RL.** We consider a Markov Decision Process (MDP), $(S, \mathcal{A}, p, r, \mu_0, \gamma)$ where $S$ is the environment space, $\mathcal{A}$ the action space, $r$ the reward distribution ($r_t \sim r(\cdot|s_t, a_t)$), $p$ is the transition probability distribution ($s_{t+1} \sim p(\cdot|s_t, a_t)$), $\mu_0$ is the initial state distribution and $\gamma \in (0, 1)$ denotes the discount factor. For the purpose of the notation, we define $p(s_0) := \mu_0(s_0)$ and we write the MDP $(S, \mathcal{A}, p, r, \gamma)$.

Actions are taking following a policy $\pi$ which depends on the environment state, (i.e $a_t \sim \pi(\cdot|s_t)$). A sequence on the MDP $(S, \mathcal{A}, p, r, \gamma)$, $\tau = s_0, a_0, r_0, \ldots s_H$, with $s_i \in S$ and $a_i \in \mathcal{A}$ is called a trajectory. A trajectory of fixed length $H \in \mathbb{N}$ is called an episode. Given a policy $\pi$, a rollout from state-action $(s, a) \in S \times \mathcal{A}$ is a random sequence $\{(s_0, a_0, r_0), (s_1, a_1, r_1), \ldots\}$ where $a_0 = a$, $s_0 = s$, $s_{t+1} \sim p(\cdot|s_t, a_t)$, $r_t \sim r(\cdot|s_t, a_t)$ and $a_t \sim \pi(\cdot|s_t)$. Given a policy $\pi$ and a fixed length $H$, we have a trajectory distribution given by

$$p_\pi(\tau) = p(s_0) \prod_{t=0}^{H-1} \pi(a_t|s_t)p(s_{t+1}|s_t, a_t)r(r_t|s_t, a_t)$$

The goal of classical risk-neutral RL algorithms is to find a policy which maximizes the expected discounted return $\mathbb{E}_\pi[\sum_{t=0}^{H} \gamma^t r_t]$ where $H$ might be infinite. Equivalently, we can look for the policy which maximizes the $Q$-function which is defined as $Q_\pi : S \times \mathcal{A} \to \mathbb{R}$, $Q_\pi(s_t, a_t) := \mathbb{E}_\pi[\sum_{t'=t}^{H} \gamma^{t'-t} r_{t'}]$. Instead of this classical objective function, other choices are possible. For example, the maximum entropy RL objective function $\mathbb{E}_\pi[\sum_{t=0}^{H} r_t + \mathcal{H}(\pi(\cdot|s_t))]$, where $\mathcal{H}$ denotes the entropy. This function has an interesting connection with variational inference (Ziebart, 2010; Levine, 2018) and it has shown impressive results in recent years (Haarnoja et al., 2018; Lee et al., 2019; Rafailov et al., 2021).

In offline RL, we have access to a fixed dataset $\mathcal{D} = \{(s_t, a_t, r_t, s_{t+1})\}$, where $s_t \in S$, $a_t \in \mathcal{A}$, $r_t \sim r(s_t, a_t)$ and $s_{t+1} \sim p(\cdot|s_t, a_t)$, and we aim to learn policies without interaction with the environment. A such dataset comes with the empirical behaviour policy $\pi_\beta(a|s) := \frac{\sum_{(s_t, a_t) \in \mathcal{D}} \mathbf{1}_{(s_t=s, a_t=a)}}{\sum_{s_t \in S} \mathbf{1}_{s_t=s}}$.

**Latent variable model.** There are different methods to learn directly from high-dimensional states (Kostrikov et al., 2020; Hafner et al., 2019a;b). But in this work, we build on the top of the framework presented in Stochastic Latent Actor-Critic (SLAC) (Lee et al., 2019). The main idea of this work is to train a latent variable model to encode the natural MDP $(S, \mathcal{A}, p, r, \gamma)$ into a latent space $(Z, \mathcal{A}, q, r, \gamma)$ and to train policies directly in this space. To achieve this, the variational distribution $q(z_{1:H}, a_{t+1:H}|s_{1:t}, a_{1:t})$ is factorized into a product of inference term $q(z_{i+1}|z_i, s_{i+1}a_i)$, latent dynamic term $q(z_{t+1}|z_t, a_t)$ and policy term $\pi(a_t|s_{1:t}, a_{1:t-1})$ as follow

$$q(z_{1:H}, a_{t+1:H}|s_{1:t}, a_{1:t}) = \prod_{i=0}^{t} q(z_{i+1}|z_i, s_{i+1}a_i) \prod_{i=t+1}^{H-1} q(z_{t+1}|z_t, a_t) \prod_{i=t+1}^{H-1} \pi(a_t|s_{1:t}, a_{1:t-1})$$

Using this factorization, the evidence lower bound (ELBO) (Odaibo, 2019) and a really interesting theoretical approach (Levine, 2018), the following objective function for the latent variable model is derived

$$\mathbb{E}_{z_{1:t}, a_{t+1:H} \sim q} \left[ \sum_{i=0}^{t} \log D(s_{t+1}|z_{t+1}) - D_{KL}(q(z_{t+1}|s_{t+1}, s_t, a_t) || q(s_{t+1}|s_t, a_t)) \right] \tag{3}$$

where $D_{KL}$ is the Kullback-Leibler divergence and $D$ a decoder.

**Distributional RL.** The goal of distributional RL is to learn the distribution of the discounted cumulative negative rewards $Z_\pi := -\sum_{t}^{H} \gamma^t r_t$. $Z_\pi$ is a random variable. A classical approach of distributional RL is to learn $Z_\pi$ implicitly using its quantile function $F_{Z(s,a)}^{-1} : [0, 1] \rightarrow \mathbb{R}$, which is defined as $F_{Z(s,a)}^{-1}(y) := \inf\{x \in \mathbb{R} \mid y \leq F_{Z(s,a)}(x)\}$ and where $F_{Z(s,a)}$ is the cumulative density function of the random variable $Z(s,a)$. A model $Q_\theta(\eta, s, a)$ is used to approximate $F_{Z(s,a)}^{-1}(\eta)$. For $(s, a, r, s') \sim \mathcal{D}$, $a' \sim \pi(\cdot|s')$, $\eta, \eta' \sim \text{Uniform}[0, 1]$, we define $\delta$ as $\delta := r + \gamma Q_{\theta'}(\eta', s', a') - Q_\theta(\eta, s, a)$. $Q_\theta$ is trained using the $\tau$-Huber quantile regression loss at threshold $k$ (Huber, 1992)

$$\mathcal{L}_k(\delta, \eta) := \begin{cases} |\eta - \mathbf{1}_{\delta<0}|(\delta^2/2k) & \text{if } |\delta| < k \\ |\eta - \mathbf{1}_{\delta<0}|(|\delta| - k/2) & \text{otherwise.} \end{cases} \tag{4}$$

With this function $F_{Z(s,a)}$, different risk measures can be computed, like Cumulative Probability Weight (CPW) (Tversky & Kahneman, 1992), Wang measure (Wang, 2000) or Conditional Value-at-Risk. For example, the following equation (Acerbi, 2002) is used to compute CVaR$_\alpha$

$$\text{CVaR}_\alpha(Z_\pi) = \frac{1}{1-\alpha} \int_\alpha^1 F_{Z^{-1}(s,a)}(\tau) d\tau \tag{5}$$

It is also possible to extend distributional RL in a offline settings. For example, O-RAAC (Urpí et al., 2021) and following a previous work (Fujimoto et al., 2019), decomposes the actor into two different component an imitation actor and a perturbation model. CODAC (Ma et al., 2021) extends DSAC (Duan et al., 2021) in a offline settings. More precisely, the $Q_\theta(\eta, s, a)$ are trained using the following loss function

$$\alpha \mathbb{E}_{\eta \sim U} \left[ \mathbb{E}_{s \sim \mathcal{D}} \left[ \log \sum_a \exp(Q_\theta(\eta, s, a)) \right] - \mathbb{E}_{(s,a) \sim \mathcal{D}} [Q_\theta(\eta, s, a)] \right] + \mathcal{L}_k(\delta, \eta') \tag{6}$$

where $U = \text{Uniform}[0, 1]$. The first term of the equation, and based on previous work (Kumar et al., 2020), is introduced to prevent for too optimistic estimation for OOD state-action pairs. The second term, $\mathcal{L}_k(\delta, \tau)$, is the classical objective function used to train the Q-function in a distributional settings.

## 4 THEORETICAL CONSIDERATIONS

In this paper, we aim to train policies in a high-dimensional stochastic and offline settings. For a practical point of view, our idea is quite straightforward. Since training with high dimensional space directly failed and following previous work (Rafailov et al., 2021; Lee et al., 2019), we encode our

high-dimensional states in more compact representation using $\phi : S \to Z$, where $Z = \phi(S)$. Using $\phi$, we build a MDP on the latent space and train policies directly on the top of this space.

However, for a theoretical point of view, this general idea is not really clear. Is this latent MDP always well defined ? If we find a policy which minimizes a risk measure in the latent space, will it also minimize the risk measure in the natural state ?

The first goal of this section is to rigorously construct a MDP in the latent space. Then, we show that for special coherent risk measure and assuming some hypothesis, minimizing the risk measure in the latent space and in the natural space is equivalent. In particular, we show that is the case for Conditional Value-at-Risk.

## 4.1 THEORETICAL RESULTS

First, we make the following assumptions

1. $\forall a \in \mathcal{A}$ we have $r(\cdot|s, a) = r(\cdot|s', a)$ if $\phi(s) = \phi(s')$.

2. We note $\mathbb{P}(\cdot|s_t, a_t)$ the probability measure with probability density function $p(\cdot|s_t, a_t)$. We suppose that if $s_t, s'_t \in S$ satisfy $\phi(s_t) = \phi(s'_t)$, then $\mathbb{P}(\cdot|s_t, a_t) = \mathbb{P}(\cdot|s'_t, a_t)$.

3. We note $\mathbb{Q}(\cdot|z_t, a_t)$ the probability image of $\mathbb{P}(\cdot|s_t, a_t)$ by $\phi$ and where $s_t$ is any element of $\phi^{-1}(z_t)$. We suppose $\mathbb{Q}(\cdot|z_t, a_t)$ admits a probability density functions $q(\cdot|z_t, a_t)$.

Under these assumptions, $\phi$ induces a MDP $(Z, \mathcal{A}, q, r', \gamma)$ where the reward distribution $r'$ is defined as $r'(\cdot|z, a) := r(\cdot|s, a)$ for any $s \in \phi^{-1}(z)$. $r'$ is well defined by surjectivity of $\phi$ and by assumption (1). $\mathbb{Q}$ is well defined by hypothesis (2).

Obviously, for a given policy $\pi'$ on the MDP $(Z, \mathcal{A}, q, r', \gamma)$ and fixed length $H$, we have a trajectory distribution

$$q_{\pi'}(\tau') = q(z_0) \prod_{t=0}^{H-1} \pi'(a_t|z_t) q(z_{t+1}|z_t, a_t) r'(r_t|z_t, a_t)$$

Given a policy $\pi$ and a fixed length $H$, we denote $\Omega$ the set of all trajectories on the MDP $(S, \mathcal{A}, p, r, \gamma)$, $\mathcal{F}$ the $\sigma$-algebra generated by these trajectories and $\mathbb{P}_\pi$ the probability with probability distribution $p_\pi$. Following the same idea and given a policy $\pi'$, we denote $\Omega'$ the set of trajectories of length $H$ on the MDP $(Z, \mathcal{A}, q, r')$, $\mathcal{F}'$ the $\sigma$-algebra generated by these trajectories and $\mathbb{Q}_{\pi'}$ the probability with probability distribution $q_{\pi'}$. $(\Omega, \mathcal{F}, \mathbb{P}_\pi)$ and $(\Omega', \mathcal{F}', \mathbb{Q}_{\pi'})$ are probability spaces.

$\phi : S \to Z$ induces a map

$$\Omega : \to \Omega'$$
$$(s_0, a_0, r_0, \ldots, s_H) \mapsto (\phi(s_0), a_0, r_0, \ldots, \phi(s_0))$$

With a slight abuse of notation, we write it $\phi$.

We denote $\Pi$ the set of all policies $\pi$ on the MDP $(S, \mathcal{A}, p, r)$ which satisfies $\pi(a|s) = \pi(a|s')$ if $\phi(s) = \phi(s')$. If $\pi \in \Pi$, then $\pi$ induces a policy $\pi'$ on the MDP $(Z, \mathcal{A}, q, r')$, taking $\pi'(a|z) := \pi(a|s)$ where $s$ is any element of $\phi^{-1}(z)$.

For any $\pi \in \Pi$ we denote $\pi'$ the associated policy on the MDP $(Z, \mathcal{A}, q, r')$ as defined above. Furthermore, we note $\Pi' := \{\pi' \mid \pi \in \Pi\}$. If $\pi' \in \Pi'$, a policy $\pi \in \Pi$ can be defined $\pi(a|s) := \pi'(a|\phi(s))$. $X$ is a random variable on $\Omega$ and $X'$ a random variable on $\Omega'$.

With all these notations, we can introduce our first result.

**Lemma 4.1.1.** *Let $\pi \in \Pi$. Then, $\mathbb{Q}_{\pi'}$ is the probability image of $\mathbb{P}_\pi$ by $\phi$.*

This result is not really surprising but it has an interesting implication. Indeed, it applies that if $X' \circ \phi = X$, then $\mathbb{E}_{q_{\pi'}}[X'] = \mathbb{E}_{p_\pi}[X]$. And thus, we get the following result.

**Corollary 4.1.2.** *Suppose $X' \circ \phi = X$. Then, if a policy $\pi'_\star$ satisfies*

$$\pi'_\star = \arg\min_{\pi' \in \Pi'} \mathbb{E}_{q_{\pi'}}[X']$$

*its associated policy $\pi_\star$ satisfies*

$$\pi_\star = \arg\min_{\pi \in \Pi} \mathbb{E}_{p_\pi}[X]$$

For a coherent risk measure $\mathcal{R}$, we write $\mathcal{U}$ the risk envelope associated to $\mathcal{R}$ in $\Omega$ and $\mathcal{U}'$ the corresponding associated risk envelope in $\Omega$. If $\forall \delta \in \mathcal{U}, \exists \delta' \in \mathcal{U}'$ such that $\delta = \delta' \circ \phi$ and $\forall \delta' \in \mathcal{U}'$ $\exists \delta \in \mathcal{U}$ with $\delta = \delta' \circ \phi$ almost everywhere, we write $\mathcal{U} = \mathcal{U}' \circ \phi$.

**Proposition 4.1.3.** *Let $\mathcal{R}$ be a coherent risk measure. Suppose that $X' \circ \phi = X$ and $\mathcal{U} = \mathcal{U}' \circ \phi$. Then, if a policy $\pi'_\star$ satisfies*

$$\pi'_\star = argmin_{\pi' \in \Pi'} \mathcal{R}(X')$$

*its associated policy $\pi_\star$ verifies*

$$\pi_\star = argmin_{\pi \in \Pi} \mathcal{R}(X)$$

This last result points out the role of the risk envelope in this equivalence.

However, the result of the last proposition is also verifies if $\sup_{\delta' \in \mathcal{U}'} \mathbb{E}_{q_{\pi'}}[\delta' X'] = \sup_{\delta \in \mathcal{U}} \mathbb{E}_{p_\pi}[\delta X]$. Fortunately, this is the case for Conditional Value-at-Risk.

**Lemma 4.1.4.** *Let $\mathcal{U}, \mathcal{U}'$ the risk envelopes associated to $CVaR_\alpha(X)$ and $CVaR_\alpha(X')$ respectively. Then, we have*

$$\sup_{\delta' \in \mathcal{U}'} \mathbb{E}_{q_{\pi'}}[\delta' X'] = \sup_{\delta \in \mathcal{U}} \mathbb{E}_{p_\pi}[\delta X]$$

Therefore, we can deduce the following result.

**Corollary 4.1.5.** *Suppose that $X' \circ \phi = X$. Then, if a policy $\pi'_\star$ satisfies*

$$\pi'_\star = argmin_{\pi' \in \Pi'} CVaR_\alpha(X')$$

*its associated policy $\pi_\star$ verifies*

$$\pi_\star = argmin_{\pi \in \Pi} CVaR_\alpha(X)$$

In particular, for $X(\tau) = \sum_{t=0}^{H} -\gamma^t r_t - \beta \mathcal{H}(\pi(\cdot|s_t))$, and $X'(\tau') := \sum_{t=0}^{H} -\gamma^t r_t - \beta \mathcal{H}(\pi'(\cdot|z_t))$, (with $\beta \in \mathbb{R}$) we have

$$(X' \circ \phi)(\tau) = \sum_{t=0}^{H} -\gamma^t r_t - \beta \mathcal{H}(\pi'(\cdot|\phi(s_t))) = \sum_{t=0}^{H} -\gamma^t r_t - \beta \mathcal{H}(\pi(\cdot|s_t)) = X(\tau)$$

Thus, we obtain the following result.

**Corollary 4.1.6.** *Let $X : \Omega \to \mathbb{R}$ and $X' : \Omega' \to \mathbb{R}$ defined as $X(\tau) := \sum_{t=0}^{H} -\gamma^t r_t - \beta \mathcal{H}(\pi(\cdot|s_t))$ and $X'(\tau') := \sum_{t=0}^{H} -\gamma^t r_t - \beta \mathcal{H}(\pi'(\cdot|z_t))$. Then, if a policy $\pi'_\star$ satisfies*

$$\pi'_\star = argmin_{\pi' \in \Pi'} CVaR_\alpha(X')$$

*its associated policy $\pi_\star$ verifies*

$$\pi_\star = argmin_{\pi \in \Pi} CVaR_\alpha(X)$$

## 4.2 DISCUSSION

The results presented in the last section provide a theoretical equivalence between minimizing the risk measure in the latent space and in the natural space. However, to obtain this guarantee we need to make some assumptions.

Assumptions (1), (2) guarantee that the reward distribution $r(\cdot|s_t, a_t)$ and the probability measure $\mathbb{P}(\cdot|s_t, a_t)$ are insensitive to any change in $\phi^{-1}(z_t)$. Moreover $\pi \in \Pi$ ensures that the policy distribution is stable to any change in $\phi^{-1}(z_t)$. Thus, and roughly speaking these assumptions guarantee we do not loss information encoding $s_t$ into $\phi(s_t)$, in term of reward distribution, transition probability measure and the optimal policy.

These theoretical considerations points out that to learn a good latent representation of the natural MDP, we should consider all components of the MDP and not only focus on the environment space.

## 5  LATENT OFFLINE DISTRIBUTIONAL ACTOR-CRITIC

The theoretical results presented above justify our really natural idea: encode the natural environment space $S$ into a smaller representation $Z$ and then use a risk-sensitive offline RL algorithm to learn a policy on the top of this space. This is the general idea of LODAC.

### 5.1  PRACTICAL IMPLEMENTATIONS OF LODAC

In this section, we present the practical implementation of LODAC. First, we need to learn a latent variational latent model. To train a such model, and similar to Rafailov et al. (2021), the following objective function is used

$$\mathbb{E}_{q_\theta}\left[\sum_{t=0}^{H-1}\log D(s_{t+1}|z_{t+1}) - D_{KL}(\phi_\theta(z_{t+1}|s_{t+1}, z_t, a_t)||q_\theta(z_{t+1}|z_t, a_t))\right] \quad (7)$$

Then, the dataset $\mathcal{D}$ is encoded into the latent space and stored into a replay buffer $\mathcal{B}_{\text{latent}}$. More precisely, $\mathcal{B}_{\text{latent}}$ contain transitions of the form $(z_{1:H}, r_{1:H}, a_{1:H})$ where $z_{1:H} \sim \phi_\theta(\cdot|s_{1:H}, a_{H-1})$ and $s_{1:H}, r_{1:H-1}, a_{1:H-1} \sim \mathcal{D}$. After that, we introduce a latent buffer $\mathcal{B}_{\text{synthetic}}$ which contains rollouts transitions performed using the policy $\pi_\theta$, the latent model $q_\theta$ and a reward estimator $r_\theta$.

The actor $\pi_\theta$ and the critic $Q_\theta$ are trained on $\mathcal{B} := \mathcal{B}_{\text{synthetic}} \cup \mathcal{B}_{\text{latent}}$. To achieve this, we could use any offline risk-sensitive RL algorithm. But based on our empirical results we follow Ma et al. (2021). Thus, the critic $Q_\theta(\eta, z_t, a_t)$ is used to approximate the inverse quantile function. $Q_\theta$ is iteratively chosen to minimize

$$\alpha\mathbb{E}_{\eta\sim U}\left[\mathbb{E}_{z\sim\mathcal{B}}\left[\log\sum_a \exp(Q_\theta(\eta, z, a))\right] - \mathbb{E}_{(z,a)\sim\mathcal{B}}[Q_\theta(\eta, z, a)]\right] + \mathcal{L}_k(\delta, \eta') \quad (8)$$

where $U = \text{Uniform}[0, 1]$, $\delta = r_t + \gamma Q_\theta(\eta', z_{t+1}, a_{t+1}) - Q_\theta(\eta, z_t, a_t)$, with $(z_t, a_t, r_t) \sim \mathcal{B}$, $a_{t+1} \sim \pi_\theta(\cdot|z_t)$ and $\mathcal{L}_k$ is the $\tau$-Huber quantile regression loss as defined is (4). The actor is trained to minimize Conditional Value-at-Risk of the negative cumulative reward, which can be computed using the formula

$$\text{CVaR}_\alpha(Z_\pi) = \frac{1}{1-\alpha}\int_\alpha^1 F_{Z^{-1}(s,a)}(\tau)d\tau \quad (9)$$

Following previous works (Rafailov et al., 2021; Yu et al., 2021b), batches of equal data mixed from $\mathcal{B}_{\text{latent}}$ and $\mathcal{B}_{\text{synthetic}}$ are used to train $\pi_\theta$ and $Q_\theta$. A summary of LODAC can be found in Appendix D.

### 5.2  EXPERIMENTAL SETUP

In this section, we evaluate the performance of LODAC.

First, our method is compared with LOMPO (Rafailov et al., 2021), which is an offline high-dimensional risk-free algorithm. Then, we build a version of LODAC where the actor and the critic are trained using O-RAAC (Urpí et al., 2021). We denote it LODAC-O. Moreover, it is also possible to use a risk-free offline RL. Thus, a risk-free policy is also trained in the latent space using COMBO (Yu et al., 2021b).

These algorithms are evaluated using the standard walker walk task from the DeepMind Control suite (Tassa et al., 2018), but here, we learn directly from the pixels. As a standard practice, each action is repeated two times on the ground environment and episode of length 1000 are used. These algorithms are tested on three different datasets : expert, medium and expert-replay. Each dataset consists of 100K transitions steps.

For the stochastic environment, we transform the reward using the following formula

$$r_t \sim \left(r(s_t, a_t) - \lambda\mathbf{1}_{\text{r}(s_t, a_t)>\bar{\text{r}}}\mathcal{B}_{p_0}\right)$$

where $r$ is the classical reward function. $\bar{r}$, $\lambda$ are hyperparameters and $\mathcal{B}_{p_0}$ is a Bernoulli distribution of parameter $p_0$. In our experiments, we choose $\lambda = 8$ and $p_0 = 0.1$. Different value of $\bar{r}$ are used

Table 1: Performance for the stochastic offline high-dimensional walker_walk task on expert, medium and expert_replay datasets. We compare the mean and the $CVaR_{0.7}$ of the returns. LODAC outperforms other algorithms on the medium and expert_replay datasets while achieving comparable results on the expert dataset. We bold the highest score across all methods.

| Dataset type | Algorithm | Mean | $CVaR_{0.7}$ |
|---|---|---|---|
| e xpert | LOMPO | 8.67 | 1.92 |
| expert | COMBO | 10.96 | 3.72 |
| expert | LODAC-O | 18.54 | **4.92** |
| expert | LODAC | **18.70** | 4.32 |
| medium | LOMPO | 19.03 | 13.31 |
| medium | COMBO | 16.25 | 11.18 |
| medium | LODAC-O | 21.31 | 14.99 |
| medium | LODAC | **27.85** | **21.37** |
| expert_replay | LOMPO | 23.22 | 16.03 |
| expert_replay | COMBO | 20.01 | 14.12 |
| expert_replay | LODAC-O | 5.73 | 0.28 |
| expert_replay | LODAC | **39.4** | **30.14** |

Table 2: Performance for the deterministic offline high-dimensional walker_walk task on expert, medium and expert_replay datasets. We compare the mean and the $CVaR_{0.7}$ of the returns. We bold the highest score across all methods.

| Dataset type | Algorithm | Mean | $CVaR_{0.7}$ |
|---|---|---|---|
| e xpert | LOMPO | 14.56 | 0.99 |
| expert | COMBO | **16.74** | 1.97 |
| expert | LODAC-O | 13.25 | **2.79** |
| expert | LODAC | 16.03 | 0.0 |
| medium | LOMPO | **52.2** | **42.56** |
| medium | COMBO | 43.71 | 25.89 |
| medium | LODAC-O | 37.33 | 31.12 |
| medium | LODAC | 33.08 | 5.79 |
| expert_replay | LOMPO | 55.86 | **35.10** |
| expert_replay | COMBO | **62.88** | 27.69 |
| expert_replay | LODAC-O | 15.81 | 2.90 |
| expert_replay | LODAC | 48.01 | 27.25 |

for each dataset such that about the half of the states verify $r(s_t, a_t) > \overline{r}$. More details regarding the construction of these datasets can be found in Appendix B.

Algorithms have been tested using the following procedure. First of all, to avoid computation time and for a more accurate comparison, we use the same latent variable model for all algorithms. We evaluate each algorithm using 100 episodes, reporting the mean and $CVaR_\alpha$ of the returns. LODAC and LODAC-O are trained to minimize $CVaR_{0.7}$. LOMPO and COMBO are trained to maximize the return. We run 4 different random seeds. As introduced in Fu et al. (2020), we use the normalized score to compare our algorithms. More precisely, a score of 0 corresponds to a fully random policy and a score of 100 corresponds to an expert policy on the deterministic task. However, and as suggested in Agarwal et al. (2021), instead of taking the mean and the results , we consider the interquartile means (IQN).

### 5.3 RESULTS DISCUSSION

In this section, we discuss the results of our experiments. Our experimental results can be read in the table 1 for the stochastic environment in table 2 for the deterministic environment. We bold the highest score across all methods. Complete results of all different tests can be found in Appendix E.

**Stochastic environment.** The first general observation is that LODAC and LODAC-O generally outperform risk-free algorithms. The only exception is with the expert_replay dataset where LODAC-O provides worse results. This is not really surprising since actors trained with O-RAAC contains an imitation component and obviously the imitation agent obtains really poor performance on this dataset. Then, it can be noticed, that LODAC provides really interesting results. Indeed it significantly outperforms risk-free algorithm in term of $CVaR_{0.7}$ and return on all datasets. Moreover, it provides better results than LODAC-O on the medium and the expert_replay dataset, while achieving comparable result on the expert dataset. A final observation is that risk-free RL policies provide generally bad performances on this stochastic environment.

**Deterministic environment.** First, a drop of performance between the deterministic and the stochastic environment can be noticed. This is not really surprising since stochastic environments are more challenging than deterministic environments. However this modification seems to affect more risk-free algorithms than LODAC-O or LODAC. Indeed, we get a difference of more than $27\%$ with LOMPO and COMBO on the medium and expert_replay dataset in term of return between the deterministic and the stochastic environment. This difference even achieves $42\%$ for COMBO on the expert_replay dataset. For LODAC-O and for the same tasks, we obtain a deterioration of less than $17\%$. In the opposite to previous risk-free methods and in a lesser degree LODAC-O, adding stochasticity in the dataset does not seem to have a big impact of the performance of LODAC. Indeed, with this algorithm we observe a drop of performance of less than $9\%$. For the medium dataset, a difference of only $5,23\%$ can even be noticed.

## 6 CONCLUSION

While offline RL appears to be an interesting paradigm for real-world application, some of these real-world application are high-dimensional and stochastic. However, current high-dimensional offline RL algorithms are trained and tested in deterministic environments. Our empirical results suggest that adding stochasticity to the training dataset significantly decreases the performance of high-dimensional risk-free offline RL algorithms.

Based on this observation, we develop LODAC. LODAC can be used to train policies in high-dimensional stochastic and offline settings. Our theoretical considerations in section 4 show that our algorithm relies on a strong theoretical foundation. In the opposite to previous risk-free methods, adding stochasticity in the dataset does not seem to have a big impact of the performance of LODAC. Finally, the use of LODAC to minimize Conditional Value-at-Risk empirically outperforms previous algorithms in term of CVaR and return on stochastic high-dimensional environments.

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

# A  PROOFS

## A.1  LEMMA 4.1.1

**Lemma A.1.1.** *Let $\pi \in \Pi$. Then, $\mathbb{Q}_{\pi'}$ is the probability image of $\mathbb{P}_\pi$ by $\phi$.*

*Proof.* Let $B' := Z_0 \times A_0 \times R_0 \times \ldots Z_H \in \mathcal{F}'$. We have

$$
\mathbb{Q}_{\pi'}(B') = \int_{B'} q_{\pi'}(\tau')d\tau'
$$

$$
= \int_{Z_0} q(z_0) \left( \int_{A_0 \times R_0 \times \ldots \times Z_H} \pi'(a_0|z_0)r'(r_0|a_0,z_0)\ldots q(z_z|z_{H-1},a_{H-1})da_0 dr_0 \ldots dz_H \right) dz_0
$$

$$
= \int_{\phi^{-1}(Z_0)} p(s_0) \left( \int_{A_0 \times R_0 \times \ldots \times Z_H} \pi'(a_0|\phi(s_0))r'(r_0|a_0,\phi(s_0))\ldots q(z_z|z_{H-1},a_{H-1})da_0 dr_0 \ldots dz_H \right) ds_0
$$

$$
= \int_{\phi^{-1}(Z_0)} p(s_0) \left( \int_{A_0 \times R_0 \times \ldots \times Z_H} \pi(a_0|s_0)r(r_0|a_0,s_0)\ldots q(z_z|z_{H-1},a_{H-1})da_0 dr_0 \ldots dz_H \right) ds_0
$$

$$
= \int_{\phi^{-1}(Z_0) \times A_0 \times R_0} p(s_0)\pi(a_0|s_0)r(r_0|a_0,s_0)
$$

$$
\left( \int_{Z_1 \times A_1 \times \ldots \times Z_H} q(z_1|\phi(s_0))\pi'(a_1|z_1)\ldots q(z_z|z_{H-1},a_{H-1})dz_1 \ldots dz_H \right) ds_0 da_0 dr_0
$$

$$
= \int_{\phi^{-1}(Z_0) \times A_0 \times R_0 \times \phi^{-1}(Z_1)} p(s_0)\pi(a_0|s_0)r(r_0|a_0,s_0)p(s_1|s_0,a_0)
$$

$$
\left( \int_{A_1 \times \ldots \times Z_H} q(z_1|\phi(s_0))\pi'(a_1|z_1)\ldots q(z_z|z_{H-1},a_{H-1})da_1 \ldots dz_H \right) ds_0 da_0 dr_0 ds_1
$$

$$
= \ldots
$$

$$
= \int_{\phi^{-1}(Z_0) \times \ldots \phi^{-1}(Z_H)} p(s_0)\pi(a_0|s_0)r(r_0|a_0,s_0)\ldots p(s_H|s_{H-1},a_{H-1})ds_0 \ldots ds_H
$$

$$
= \int_{\phi^{-1}(B')} p_\pi(\tau)d\tau
$$

$$
= \mathbb{P}_\pi(\phi^{-1}(B'))
$$

$\square$

## A.2  PROPOSITION 4.1.3

**Proposition A.2.1.** *Let $\mathcal{R}$ be a coherent risk measure. Suppose that $X' \circ \phi = X$ and $\mathcal{U} = \mathcal{U}' \circ \phi$. Then, if a policy $\pi'_\star$ satisfies*

$$
\pi'_\star = argmin_{\pi' \in \Pi'} \mathcal{R}(X')
$$

*its associated policy $\pi_\star$ verifies*

$$
\pi_\star = argmin_{\pi \in \Pi} \mathcal{R}(X)
$$

*Proof.* Since $\mathcal{U} = \mathcal{U}' \circ \phi$, we have

$$
\sup_{\delta' \in \mathcal{U}'} \mathbb{E}_{q_{\pi'}}[\delta' X'] = \sup_{\delta \in \mathcal{U}} \mathbb{E}_{p_\pi}[\delta X]
$$

Thus

$$
\mathcal{R}_{q_{\pi'}}(X') := \sup_{\delta' \in \mathcal{U}'} \mathbb{E}_{q_{\pi'}}[\delta' X'] = \sup_{\delta \in \mathcal{U}} \mathbb{E}_{p_\pi}[\delta X] = \mathcal{R}_{p_\pi}(X)
$$

Now, let $\pi'_\star = argmin_{\pi' \in \Pi} \mathcal{R}_{q_{\pi'}}(X')$ and $\pi_\star$ the associated policy in $S$. By contradiction, suppose there exists $\pi_1$ with $\mathcal{R}_{p_{\pi_1}}(X) < \mathcal{R}_{p_{\pi_\star}}(X)$. But in this case and by the above observation, we would have $\mathcal{R}_{q_{\pi'_1}}(X') < \mathcal{R}_{q_{\pi'_\star}}(X')$. $\square$

### A.3 Lemma 4.1.4

**Lemma A.3.1.** *Let $\mathcal{U}$, $\mathcal{U}'$ the risk envelopes associated to $CVaR_\alpha(X)$ and $CVaR_\alpha(X')$ respectively. Then, we have*

$$\sup_{\delta' \in \mathcal{U}'} \mathbb{E}_{q_{\pi'}}[\delta' X'] = \sup_{\delta \in \mathcal{U}} \mathbb{E}_{p_\pi}[\delta X]$$

*Proof.* Recall that the risk envelope of the $CVaR_\alpha$, takes the form $\mathcal{U} = \{\delta \mid 0 \leq \delta \leq \frac{1}{\alpha}, \mathbb{E}[\delta] = 1\}$.

- Let $\delta' \in \mathcal{U}'$. We define $\delta := \delta' \circ \phi$. By construction we have $0 \leq \delta \leq \frac{1}{\alpha}$, $\mathbb{E}_{p_\pi}[\delta] = \mathbb{E}_{q_{\pi'}}[\delta'] = 1$ and $\mathbb{E}_{q_{\pi'}}[\delta' X'] = \mathbb{E}_{p_\pi}[\delta X]$.

- Let $\delta \in \mathcal{U}$. By definition, $\delta p_\pi$ is a density function. Let $\tilde{\mathbb{P}}_\pi$ its associated probability measure. We consider $\tilde{\mathbb{Q}}_{\pi'}$ the probability image of $\tilde{\mathbb{P}}_\pi$ by $\phi$. Then, remark that $\tilde{\mathbb{Q}}_{\pi'}$ is absolutely continuous with respect to $\mathbb{Q}_{\pi'}$. Indeed, if $A \in \mathcal{F}'$ satisfies $\mathbb{Q}_{\pi'}(A) = 0$, we have

$$\tilde{\mathbb{Q}}_{\pi'}(A) = \tilde{\mathbb{P}}_\pi(\phi^{-1}(A)) = \int_{\phi^{-1}(A)} \delta(\tau) p_\pi(\tau) d\tau$$

$$\leq \frac{1}{\alpha} \mathbb{P}_\pi(\phi^{-1}(A)) = \frac{1}{\alpha} \mathbb{Q}_{\pi'}(A) = 0$$

Thus by Radon-Nikodym, there exists $\delta' \geq 0$, such that for all $B \in \mathcal{F}'$

$$\tilde{\mathbb{Q}}_{\pi'}(B) = \int_B \delta' d\mathbb{Q}_{\pi'} = \int_B \delta'(\tau') q_{\pi'}(\tau') d\tau'$$

We will show that à $\delta' \in \mathcal{U}'$. We define $C := \{\tau' \in \Omega' \mid \delta'(\tau) > \frac{1}{\alpha}\}$. By contradiction, suppose that $\mathbb{Q}_{\pi'}(C) > 0$. One one hand have

$$\tilde{\mathbb{Q}}_{\pi'}(C) = \int_C \delta'(\tau') q_{\pi'}(\tau') d\tau' > \frac{1}{\alpha} \int_C q_{\pi'}(\tau') d\tau' = \frac{1}{\alpha} \mathbb{Q}_{\pi'}(C)$$

And the other hand, we have

$$\tilde{\mathbb{Q}}_\pi(C) = \tilde{\mathbb{P}}_\pi(\phi^{-1}(C)) = \int_{\phi^{-1}(C)} \delta(\tau) p_\pi(\tau) d\tau \leq \frac{1}{\alpha} \mathbb{P}_\pi(\phi^{-1}(C)) = \frac{1}{\alpha} \mathbb{Q}_{\pi'}(C)$$

And thus, we must have $\mathbb{Q}_{\pi'}(C) = 0$.

Thus, $0 \leq \delta' \leq \frac{1}{\alpha}$ a.e. and therefore $\delta' \in \mathcal{U}'$. And since, $\delta' q_{\pi'}$ is the density function of the probability image of $\tilde{\mathbb{P}}_\pi$, we obtain $\mathbb{E}_{\delta p_\pi}[X] = \mathbb{E}_{\delta' q_{\pi'}}[X']$.

$\square$

## B Datasets

In this section, we describe more precisely how we build our training datasets.

- **Expert.** For the expert dataset, actions are chosen according to an expert policy which has been trained online, in a risk-free environment using SAC for 500K training steps. For this training, the states are the classical states provided by DeepMind Control suite.

- **Medium.** In this dataset, actions are chosen according to a policy which has been trained using the same method as above, but here, the training has been stopped when it achieves about the half performance of the expert policy.

- **Expert_replay.** The expert_replay dataset consists of episode which are sampled from the expert policy during the training.

The setup presented above allow to build the deterministic datasets.

For the stochastic environment, we transform the reward using the following formula

$$r_t \sim \left( r(s_t, a_t) - \lambda \mathbf{1}_{\mathrm{r}(\mathrm{s_t, a_t}) > \bar{\mathrm{r}}} \mathcal{B}_{p_0} \right)$$

where $r$ is the classical reward function. $\bar{r}, \lambda$ are hyperparameters and $\mathcal{B}_{p_0}$ is a Bernoulli distribution of parameter $p_0$. In our experiments, we choose $\lambda = 8$ and $p_0 = 0.1$. Different value of $\bar{r}$ are used for each dataset such that about the half of the states verify $r(s_t, a_t) > \bar{r}$.

## C    IMPLEMENTATIONS DETAILS

In this section, we present some details related to our implementations.

**Latent variable model.** Following previous works (Rafailov et al., 2021; Yu et al., 2021b), our latent variable model contains the following components

$$
\begin{aligned}
\text{Image encoder :} \qquad & h_t = E_\theta(s_t) \\
\text{Inference model :} \qquad & z_t \sim \phi_\theta(\cdot | h_t, z_{t-1}, a_{t-1}) \\
\text{Latent transition model :} \qquad & z_t \sim q_\theta(\cdot | z_{t-1}, a_{t-1}) \\
\text{Reward estimator :} \qquad & r_t \sim r_\theta(\cdot | s_t) \\
\text{Image decoder :} \qquad & s_t \sim D_\theta(\cdot | z_t)
\end{aligned}
$$

The image encoder $E_\theta$ is a classical Convolutional Neural Network. The inference and the latent transition model are implemented as a Recurrent State Space Model (RSSM), (Hafner et al., 2019b). More precisely, a latent state $z_t$ has two components $z_t = [d_t, x_t]$. $d_t$ is the deterministic part and $x_t$ the stochastic. $d_t$ is computed using a GRU cell $d_t = f_\theta(z_{t-1}, a_{t-1})$ and the stochastic component $x_t$ is computed using, $x_t \sim \phi_\theta(\cdot | h_t, d_t)$, for the inference model, and $x_t \sim q_\theta(\cdot | d_t, x_{t-1}, a_{t-1})$ for the latent transition model. Both $q_\theta$ and the reward estimator are implemented as MLP. Finally, the decoder $D_\theta$ is a Deconvolutional Neural Network.

**Actor-critic.** First, recall that our initial critic loss function is

$$\alpha \, \mathbb{E}_{\eta \sim U} \left[ \mathbb{E}_{z \sim \mathcal{B}} \left[ \log \sum_a \exp(Q_\theta(\eta, z, a)) \right] - \mathbb{E}_{(z,a) \sim \mathcal{B}} \left[ Q_\theta(\eta, z, a) \right] \right] + \mathcal{L}_k(\delta, \eta') \qquad (10)$$

But following Kumar et al. (2020); Ma et al. (2021), we add two parameters $\zeta, \omega \in \mathbb{R}_{>0}$ to the last equation

$$\max_{\alpha \geq 0} \quad \alpha \, \mathbb{E}_{\eta \sim U} \left[ \omega \left[ \mathbb{E}_{z \sim \mathcal{B}} \left[ \log \sum_a \exp(Q_\theta(\eta, z, a)) \right] - \mathbb{E}_{(z,a) \sim \mathcal{B}} \left[ Q_\theta(\eta, z, a) \right] \right] - \zeta \right] + \mathcal{L}_k(\delta, \eta')$$

$\zeta$ thresholds the difference between $\mathbb{E}_{(z,a) \sim \mathcal{B}} \left[ Q_\theta(\eta, z, a) \right]$ and the regulizer $\mathbb{E}_{z \sim \mathcal{B}} \left[ \log \sum_a \exp(Q_\theta(\eta, z, a)) \right]$. The parameter $\omega$ scales this difference.

Remark that if this difference (scaled to $\omega$) is smaller than the parameter $\zeta$, then $\alpha$ will be set to 0 and only the term $\mathcal{L}_k(\delta, \eta')$ will be taken into account.

Moreover, since the action space $\mathcal{A}$ is continuous, the computation $\log \sum_a \exp(Q_\theta(\eta, z, a)$ is intractable. To avoid this problem and as introduced in (Kumar et al. (2020)) we use

$$\log \sum_a \exp(Q_\theta(\eta, z, a)) \approx \log \left( \frac{1}{2M} \sum_{a_i \sim U(\mathcal{A})}^{M} \left[ \frac{\exp(Q_\theta(\eta, z, a))}{U(\mathcal{A})} \right] + \frac{1}{2M} \sum_{a_i \sim \pi(\cdot | z)}^{M} \left[ \frac{\exp(Q_\theta(\eta, z, a))}{\pi(a_i | z)} \right] \right)$$

where $U(\mathcal{A}) = \text{Unif}(\mathcal{A})$ and we choose $M = 10$.

Then, we use an implicit quantile network (IQN), (Dabney et al., 2018) to represent our critic $Q_\theta(\eta, z, a)$ and the actor $\pi_\theta(\cdot | z)$ consists of a simple MLP which is trained to minimize

$$\text{CVaR}_\alpha(Z_\pi) = \frac{1}{1 - \alpha} \int_\alpha^1 Q_\theta(\eta, z, a) d\eta \qquad (11)$$

# D    MAIN ALGORITHM

The full implementation of LODAC can be read in algorithm (1).

---

**Algorithm 1:** LODAC

---

**Input:** dataset $\mathcal{D}$, models train steps, initial latent steps, number iterations, number actor-critic steps, rollout length L

1 **for** *models train steps* **do**
2      *Sample a batch of sequence $(s_{1:H}, a_{1:H-1}, r_{1:H-1})$ from $\mathcal{D}$ and train a variational latent model using equation (7) and a reward estimator $r_\theta$.*
3 **end**
4 **for** *Initial latent step* **do**
5      *Sample a batch of sequence $(s_{1:H}, a_{1:H-1}, r_{1:H-1})$ from $\mathcal{D}$*
6      *Sample $z_{1:H}$ from the latent model and add the transitions $(z_{1:H}, a_{1:H-1}r_{1:H-1})$ to $\mathcal{B}_{latent}$.*
7 **end**
8 **for** *initial synthetic latent steps* **do**
9      *Sample a batch of sequences $(s_{1:H}, a_{1:H-1}, r_{H-1})$ from $\mathcal{D}$*
10     *Sample a set of latent states $S \sim \phi_\theta(\cdot|s_{1:H}, a_{1:H-1})$ from the trained latent model.* **for** $s_0 \in S$ **do**
11        **for** $h \in \{1 : L\}$ **do**
12           *Sample a random action $a_{h-1}$ and $z_h \sim q_\theta(\cdot|z_{h-1}, a_{h-1})$; estimate the reward $r_t \sim r_\theta$*
13           *Add $(z_{h-1}, a_{h-1}, z_h, r_h)$ to $\mathcal{B}_{synthetic}$.*
14        **end**
15     **end**
16 **end**
17 **for** *number iterations* **do**
18     **for** *number actor-critic steps* **do**
19        *Sample transitions from $\mathcal{B}_{latent} \cup \mathcal{B}_{synthetic}$*
20        *Train the critic to minimize (8)*
21        *Train the actor to minimize $CVaR_\alpha(Z_\pi)$ which can be computed using the $Q(\eta, z, a)$ and equation (9).*
22     **end**
23     **for** *rollout steps* **do**
24        *Sample a batch of sequences $(s_{1:H}, a_{1:H-1}, r_{H-1})$ from $\mathcal{D}$*
25        *Sample a set of latent states $S \sim \phi_\theta(\cdot|s_{1:H}, a_{1:H-1})$ from the trained latent model.* **for** $s_0 \in S$ **do**
26           **for** $h \in \{1 : L\}$ **do**
27              *Sample action $a_{h-1} \sim \pi_\theta(\cdot|z_{h-1})$ and $z_h \sim q_\theta(\cdot|z_{h-1}, a_{h-1})$; estimate the reward $r_t$*
28              *Add $(z_{h-1}, a_{h-1}, z_h, r_h)$ to $\mathcal{B}_{synthetic}$.*
29           **end**
30        **end**
31     **end**
32 **end**

---

# E    COMPLETE EXPERIMENTAL RESULTS

In this section, we present our complete empirical results. More precisely, in the table 4, all results of the stochastic environment can be found. And on the table 3, all results on the deterministic can be read.

Table 3: Complete results for the deterministic offline high-dimensional walker_walk task on expert, medium and expert_replay datasets. We compare the mean and the $CVaR_{0.7}$ of the returns.

|  | Expert | | Medium | | Expert_replay | |
| --- | --- | --- | --- | --- | --- | --- |
|  | Mean | $CVaR_{0.7}$ | Mean | $CVaR_{0.7}$ | Mean | $CVaR_{0.7}$ |
| LOMPO | 174 | 65 | 419 | 298 | 410 | 274 |
|  | 177 | 98 | 583 | 501 | 523 | 341 |
|  | 71 | 26 | 515 | 437 | 601 | 411 |
|  | 193 | 37 | 527 | 428 | 586 | 387 |
| COMBO | 217 | 62 | 345 | 263 | 662 | 395 |
|  | 190 | 58 | 458 | 296 | 742 | 501 |
|  | 201 | 95 | 554 | 476 | 576 | 197 |
|  | 38 | 21 | 428 | 246 | 359 | 160 |
| LODAC-O | 149 | 68 | 385 | 334 | 107 | 59 |
|  | 178 | 44 | 428 | 357 | 194 | 104 |
|  | 149 | 73 | 381 | 309 | 224 | 51 |
|  | 186 | 67 | 384 | 321 | 180 | 78 |
| LODAC | 265 | 43 | 628 | 458 | 460 | 271 |
|  | 103 | 31 | 416 | 52 | 505 | 313 |
|  | 306 | 65 | 275 | 134 | 421 | 193 |
|  | 113 | 34 | 194 | 56 | 509 | 313 |

Table 4: Complete results for the stochastic offline high-dimensional walker_walk task on expert, medium and expert_replay datasets. We compare the mean and the $CVaR_{0.7}$ of the returns.

|  | Expert | | Medium | | Expert_replay | |
| --- | --- | --- | --- | --- | --- | --- |
|  | Mean | $CVaR_{0.7}$ | Mean | $CVaR_{0.7}$ | Mean | $CVaR_{0.7}$ |
| LOMPO | 150 | 62 | 219 | 161 | 258 | 187 |
|  | 129 | 57 | 227 | 167 | 259 | 201 |
|  | 114 | 62 | 201 | 141 | 252 | 191 |
|  | 108 | 46 | 214 | 171 | 238 | 173 |
| COMBO | 169 | 123 | 196 | 153 | 238 | 182 |
|  | 167 | 101 | 202 | 159 | 171 | 101 |
|  | 73 | 40 | 179 | 132 | 250 | 192 |
|  | 118 | 51 | 186 | 136 | 213 | 161 |
| LODAC-O | 210 | 110 | 238 | 181 | 60 | 44 |
|  | 173 | 69 | 241 | 178 | 97 | 43 |
|  | 214 | 105 | 237 | 190 | 92 | 51 |
|  | 228 | 69 | 208 | 153 | 118 | 45 |
| LODAC | 303 | 93 | 299 | 244 | 406 | 326 |
|  | 112 | 23 | 287 | 235 | 391 | 279 |
|  | 227 | 87 | 299 | 241 | 413 | 336 |
|  | 200 | 76 | 296 | 228 | 401 | 311 |

