# OpenReview forum: "Latent Offline Distributional Actor-Critic"
_ICLR.cc/2023/Conference — Submitted to ICLR 2023_

### Official Review · Reviewer_1ct4 · 2022-10-23

**Confidence:** 3
**Correctness:** 3
**Technical Novelty And Significance:** 2
**Empirical Novelty And Significance:** 2
**Recommendation:** 5

**Clarity, Quality, Novelty And Reproducibility:**

Clarity: good
Quality: good
Novelty: poor. the idea is incremental
Reproducibility: good

**Strength And Weaknesses:**

Strengths: easy to follow, good performance
Weaknesses: Incremental idea, insufficient experiments

**Summary Of The Paper:**

This paper considers the risk measure instead of the classically expected return in the offline RL setting. This paper shows that under some assumptions it is equivalent to minimize a risk measure in the latent space and in the natural space. A practical approach is proposed to use CVaR as the objective function.

**Summary Of The Review:**

The paper is easy to follow and combines distributional RL with offline RL. My concerns are:
1. the idea is incremental and the performance is not surprising. The method simply combines CVaR and an RL method. There is no specific design for the offline setting. Since distributional RL methods like C51 can achieve better performance than classical RL methods, the performance shown in this paper is not surprising.
2. For experiments, only one environment is used, which is not very convincing. More results should be provided.

typos: the equation above Eq. 3: s_i+1a_i -> s_i+1, a_i; a MDP -> an MDP; equation before Lemma 4.1.1: \phi(s_0) -> \phi(s_H)

---

> ### Author Response · Authors · 2022-11-18
> **Reply to reviewer 1ct4**
>
> > 1. the idea is incremental and the performance is not surprising. The method simply combines CVaR and an RL method. There is no specific design for the offline setting. Since distributional RL methods like C51 can achieve better performance than classical RL methods,
> > the performance shown in this paper is not surprising.
>
> Although our algorithm is a simple combination of distributional offline RL and high dimensional RL methods, we believe our paper is interesting for the following points.
>
> * First, our experiments suggest that previous high dimensional offline RL methods are not robust: adding noises in the reward function can dramatically decrease the performance of these algorithms (see Table 2 vs Table 1).
>
> * Then, we empirically show that combining high dimensional and offline distributional RL algorithms
>   provide a more robust method even if we are only interested in the expected return (see Table 1 vs Table 2).
>
> * Finally, we show that combining high dimensional and offline RL methods can be theoretically justified.
>
> > 2. For experiments, only one environment is used, which is not very convincing. More results should be provided.
>
> Working with high dimensional and distributional RL methods requires a big computation time. To keep this time reasonable, we choose to focus on only one environment.
>
> > typos: the equation above Eq. 3: s_i+1a_i -> s_i+1, a_i; a MDP -> an MDP; equation before Lemma 4.1.1: \phi(s_0) -> \phi(s_H)
>
> Thank you for pointing this out.

---

### Official Review · Reviewer_eHQL · 2022-10-24

**Confidence:** 4
**Correctness:** 3
**Technical Novelty And Significance:** 2
**Empirical Novelty And Significance:** 2
**Recommendation:** 3

**Clarity, Quality, Novelty And Reproducibility:**

The writing is coarse and the originality is limited. Code is provided for reproducibility.

**Strength And Weaknesses:**

Strengths:
1. This paper combines the methods from prior methods.

Weaknesses:
1. This paper lacks novelty. The latent model for encoding high-dimensional states is borrowed from Rafailov et al. (2021), and the distributional Q-learning is originated from Ma et al. (2021).
2. The writing is coarse, and many typos appear abruptly.
3. Theoretical analysis doesn’t support the proposed algorithm due to unsatisfactory assumptions.
4. Experiments are only conducted on customized walker datasets.

Rafael Rafailov, Tianhe Yu, Aravind Rajeswaran, and Chelsea Finn. Offline reinforcement learning from images with latent space models. In Learning for Dynamics and Control, pp. 1154–1168. PMLR, 2021.

Yecheng Ma, Dinesh Jayaraman, and Osbert Bastani. Conservative offline distributional reinforcement learning. Advances in Neural Information Processing Systems, 34, 2021.


**Summary Of The Paper:**

This paper proposes to train a distributional actor-critic algorithm in a latent space for high-dimensional data in offline reinforcement learning. Risk measure instead of the classic expected return is optimized under the risk-sensitive RL framework.

**Summary Of The Review:**

Overall, I am concerned that the originality and depth of this contribution may not quite yet reach the high bar set for ICLR. The writing errors make reading stumble.

Writing errors:
- Section 1, paragraph 2, “it may be be more …”
- Section 3, paragraph - Latent variable model, “Using this factorization, the evidence lower bound (ELBO) (Odaibo, 2019) and a really interesting theoretical approach (Levine, 2018), the following objective function for the latent variable model is derived … ” by any reference?
- Section 3, paragraph - Latent variable model, “ … and D a decoder”
- Section 3, paragraph – Distributional RL, “For example, O-RAAC (Urp´ı et al., 2021) and following a previous work (Fujimoto et al., 2019), decomposes the actor into two different component an imitation actor and a perturbation model.”
- Section 3, paragraph – Distributional RL, “… , is introduced to prevent for too optimistic estimation”
- Section 4, starting paragraph, “Since training with high dimensional space directly failed and following previous work (Rafailov et al., 2021; Lee et al., 2019), we encode our …”
- ……

Theoretical concern: the first two assumptions require the encoder $\phi$ to be a one-to-one mapping function, while this is hard to be satisfied when using non-linear neural networks state encoders, for example, $\phi$ occasionally maps both two states $s$ and $s’$ to zero vectors.  This conflicts with the use of trained variational encoder in the algorithm part, so the theoretical analysis doesn’t support the proposed algorithm.

Experiments are not standard and only walker_walk task is considered. The dataset is collected by running SAC based on classic states provided by DeepMind Control suite. If the dataset is collected by policy which is trained not based on pixel input, why you trained your model on this dataset via pixel input. Limited number of baselines are considered here, and the paper didn’t include implementation details of these baseline methods.

Is LODAC-O a O-RAAC trained with your latent encoder?

---

> ### Author Response · Authors · 2022-11-18
> **Reply to reviewer eHQL**
>
> > 1. This paper lacks novelty. The latent model for encoding high-dimensional states is borrowed from Rafailov et al. (2021), and the distributional Q-learning is originated from Ma et al. (2021).
>
> The novelty of this paper are the following.
> First, our experiments suggest that previous high dimensional offline RL methods are not robust: adding noises in the reward function can dramatically decrease the performance of these algorithms (see Table 1 vs Table 2).
> Our work suggests that combining high dimensional RL and distributional offline RL methods can prevent for this problem.
> Finally, we propose a theoretical justification of this idea.
>
> > 2. The writing is coarse, and many typos appear abruptly.
>
> We will correct the typos and try to improve the writing.
>
> > 3. Theoretical analysis doesn’t support the proposed algorithm due to unsatisfactory assumptions.
>
> Thank you for your remark. We will work on this problem.
>
> > 4. Experiments are only conducted on customized walker datasets.
>
> Working with high dimensional and distributional RL methods requires a big computation time. To keep this time reasonable, we choose to focus on only one environment.
>
> >  If the dataset is collected by policy which is trained not based on pixel input, why you trained your model on this dataset via pixel input.
>
> We do not understand why the dataset should be collected by a policy trained on pixel input. We built our training datasets using the same procedure as in some previous works (LOMPO, COMBO, ...).

---

### Official Review · Reviewer_c8jd · 2022-10-24

**Confidence:** 4
**Correctness:** 2
**Technical Novelty And Significance:** 2
**Empirical Novelty And Significance:** 1
**Recommendation:** 3

**Clarity, Quality, Novelty And Reproducibility:**

The motivations of the paper re interesting and important. However insufficient details are provided about the practical algorithm and experiments.

**Strength And Weaknesses:**

Strengths

1. The proposed approach for risk-sensitive RL in a latent space is simple to implement, by combining with several prior offline RL algorithms.
2. The motivation for necessity of modeling risk in stochastic environments is on point, and should be helpful in developing algorithms for these environments broadly.

Weaknesses

1. The practical implementation section is very vague and it is unclear how exactly the components of LODAC differ from prior approaches like Rafailov et al. (2021) More details regarding the algorithm is essential for better understanding the contributions

2. The experimental settings do not conform to the motivations for risk-sensitive RL. The only stochasticity induced is a form of Bernoulli noise added to the rewards, and the dynamics are still deterministic. I think it is important to perform analyses on environments with stochastic dynamics to really understand the benefits of risk-sensitive algorithms.

3. The experimental comparisons are insufficient and are significantly minimal compared to evaluations that offline RL algorithms usually do. For example, only a single walker agent is considered for evaluation. I would suggest at least evaluating on the entire D4RL suite and comapring with other baselines like CQL, IQL, COMBO etc.

**Summary Of The Paper:**

This paper proposes an approach for Offline RL by explicitly modeling risk through CVaR. The paper targets stochastic environments and makes the case that modeling risk is essential for  performing well in these settings by demonstrating improved performance in standard control tasks with noise injection in the rewards.

**Summary Of The Review:**

The paper has interesting insights and motivations, but the experimental comparisons are incomplete and missing external baselines, evaluation environments and experiments settings. Please refer to the Weaknesses section above.

--- AFTER REBUTTAL ---

After reading the authors' responses, I am keeping my score and not recommending accept for the paper because only my first concern is reasonably addressed, and I am not convinced by the justification for significantly minimal experimental evaluations.

---

> ### Author Response · Authors · 2022-11-18
> **Reply to reviewer c8jd**
>
>
> > 1. The practical implementation section is very vague and it is unclear how exactly the components of LODAC differ
>      from prior approaches like Rafailov et al. (2021) More details regarding the algorithm is essential for better understanding the contributions
>
> In LOMPO a recurrent variational autoencoder is used to encode the natural state into the latent space. Then, actor and the critic are trained using SAC on the top of this latent representation. For each batch half of the data comes from the latent encoded dataset and the other half from a rollout provided by the current trained agent. During this rollout, to prevent for OOD errors, the estimated rewards are penalized using a heuristic.  In LODAC, we use the same recurrent variational autoencoder. For each batch half of the data comes from a rollout provided by the current trained latent actor and the other half comes from the encoded training dataset. However, unlike in LOMPO, we did not penalize the reward estimation during the rollout.  Then, our actor and critic can be trained using any offline distributional methods and considering CVaR instead of the expected return.
> In our paper, we tested CODAC (which leads to the algorithm called LODAC) and O-RAAC (LODAC-O).
>
> > 2. The experimental settings do not conform to the motivations for risk-sensitive RL. The only stochasticity induced is a form of Bernoulli noise added to the rewards, and the dynamics are still deterministic. I think it is important to perform analyses on environments with stochastic dynamics to really understand the benefits of
> risk-sensitive algorithms.
>
> We designed our experiments following previous works in risk-sensitive RL. Indeed, it would be interesting to perform analyses on environment with stochastic dynamics. Thank you for pointing this out.
>
> > 3. The experimental comparisons are insufficient and are significantly minimal compared to evaluations that offline RL algorithms usually do. For example, only a single walker agent is considered for evaluation. I would suggest at least evaluating on the entire D4RL suite and comapring with other baselines like CQL, IQL, COMBO etc
>
> Working with high dimensional and distributional RL methods requires a big computation time. To keep this time reasonable, we choose to focus on only one environment. For the same reason we compare our method with only a few algorithms.

---

### Official Review · Reviewer_5Q9C · 2022-10-27

**Confidence:** 4
**Correctness:** 3
**Technical Novelty And Significance:** 2
**Empirical Novelty And Significance:** 2
**Recommendation:** 3

**Clarity, Quality, Novelty And Reproducibility:**

Clarity: Yes.

Quality: Yes.

Novelty: little.

Reproducibility: I believe it should be fine.

**Strength And Weaknesses:**

Strength:
 - The equivalence between minimizing risk measure in the latent space and in the natural space seems interesting.

Weaknesses:
 - Third paragraph in the intro: I don’t think I buy the logic here. “Classical” RL methods can also be evaluated under “stochastic” environment since their goal is to maximize **expected** total reward. The importance of developing risk-sensitive methods is to consider risk while maximizing total reward, which should not be counted toward training policies under stochastic environment.
 - The paper seems to be a naive combination of risk-sensitive offline RL and latent offline RL. I did not find any novelty beyond these two points. I suggest the authors further classify this.
 - Experiment seems too simple. I believe walker environment itself is already a low-dim env (18, see (Tassa et al., 2018)). It seems there is no need to further embed into low dimension. I suggest the authors consider a more complex environment.
 - The authors should also compare the results with other SOTA offline RL methods: COMBO, CQL, etc. There should be a trade-off between reward and risk.

**Summary Of The Paper:**

The paper introduces LODAC, a variant offline actor critic method, which first embeds the state space into a latent space and also include risk measure into consideration. On the theory side, the authors show that minimizing risk measure in the latent space and in the natural space. On the empirical side, the authors consider an artificial environemnt (walker) which forces the deterministic reward function to be stochastic and justifies the efficacy of the proposed method.

**Summary Of The Review:**

The paper has an overall good presentation and some motivation. However, given the weaknesses stated above, I recommend reject.

---

> ### Author Response · Authors · 2022-11-18
> **Reply to reviewer 5Q9C**
>
> > Third paragraph in the intro: I don’t think I buy the logic here.
> > “Classical” RL methods can also be evaluated under “stochastic” environment since their goal is to maximize expected total reward.
> > The importance of developing risk-sensitive methods is to consider risk while maximizing total reward, which should not be counted toward training policies under stochastic environment.
>
> Indeed, "classical RL" methods can also be evaluated under “stochastic” environment. However, our experiments suggest that classical RL methods provide bad results under high dimensional and stochastic environment, even if we consider the expected total rewards (see Table 2). We will clarify the third paragraph in the introduction.
>
> > The paper seems to be a naive combination of risk-sensitive offline RL and latent offline RL. I did not find any novelty beyond these two points.
> > I suggest the authors further classify this.
>
> Although our algorithm is a simple combination of offline distributional RL and latent offline RL algorithm, we believe our paper is interesting for the following points.
>
> * First, our experiments suggest that previous high dimensional offline RL methods are not robust: adding noises in
>   the reward function can dramatically decrease the performance of these algorithms (see Table 2 vs Table 2).
>
> * Then, we empirically show that combining high dimensional and offline distributional RL algorithms
>   provide a more robust method even if we are only interested in the expected return (see Table 2 vs Table 1).
>
> * Finally, we show that combining high dimensional and offline RL methods can be theoretically justified.
>
> > Experiment seems too simple. I believe walker environment itself is already a low-dim env (18, see (Tassa et al., 2018)). It seems there is no need to further embed into low dimension.
> > I suggest the authors consider a more complex environment.
>
> Although it might be not enough clear in our paper, the states which can be found in our datasets are the 64x64x3 pixels representation of the walker walk environment. We will clarify the second paragraph in section 5.2.
>
> > 4. The authors should also compare the results with other SOTA offline RL methods: COMBO, CQL, etc.
> > There should be a trade-off between reward and risk.
>
> Working with high dimensional and distributional RL methods require a big computation time.
> To keep this time reasonable, we choose to compare our method with only a few algorithms.

---

### Decision · Program_Chairs · 2023-01-20

**Decision:**

Reject

**Justification For Why Not Higher Score:**

The reviewers unanimously agreed to reject this paper.

**Justification For Why Not Lower Score:**

There are no lower score.

**Metareview: Summary, Strengths And Weaknesses:**

## Summary
This paper proposes a method that embeds high dimensional state space into a low dimensional latent space with a method they call as LODAC. which is a type of offline actor critic algorithm. The paper also suggests to minimize a risk measure instead of expected total rewards. The proposed method works with stochastic reward functions as well. The paper shows some empirical results on `walker_walk` task from DeepMind Control suite using pixel observations.

In a nutshell, the authors agreed that this paper is not ready for publication yet. Below I will list some of the strengths and weaknesses pointed out by the reviewers.

## Strengths

- Interesting relationship between minimizing risk measure in the latent space and in the natural space.
- The proposed approach is simple and easy to implement.

## Weaknesses

- The reviewers agreed that the proposed method is not novel and the results are not surprising.
- Limited experiments only on walker task. It is not clear how general this approach is since there are no experiments on other environments.
- Lack of comparisons against other SOTA offline RL tasks.
- Unclear and vague writing. For instance, the intro of the paper needs some work. In general, th writing is coarse, and there are several typos in the paper.
- Lack of experiments on environments with stochastic dynamics.
- Theoretical analysis doesn’t support the proposed algorithm due to unsatisfactory assumptions.